# Switching Linear Dynamics for Variational Bayes Filtering

## Abstract

System identification of complex and nonlinear systems is a central problem for model predictive control and model-based reinforcement learning. Despite their complexity, such systems can often be approximated well by a set of linear dynamical systems if broken into appropriate subsequences. This mechanism not only helps us find good approximations of dynamics, but also gives us deeper insight into the underlying system. Leveraging Bayesian inference and Variational Autoencoders, we show how to learn a richer and more meaningful state space, e.g. encoding joint constraints and collisions with walls in a maze, from partial and high-dimensional observations. This representation translates into a gain of accuracy of the learned dynamics which we showcase on various simulated tasks.

## 1 Introduction

Learning dynamics from raw data (also known as system identification) is a key component of model predictive control and model-based reinforcement learning. Problematically, environments of interest often give rise to very complex and highly nonlinear dynamics which are seemingly difficult to approximate. However, switching linear dynamical systems (SLDS) approaches claim that those environments can often be broken down into simpler units made up of areas of equal and linear dynamics (Ackerson & Fu, 1970; Chang & Athans, 1978). Not only are those approaches capable of good predictive performance, which often is the sole goal of learning a system's dynamics, they also encode valuable information into so called *switching variables* which determine the dynamics of the next transition. For example, when looking at the movement of an arm, one is intuitively aware of certain restrictions of possible movements, e.g. constraints to the movement due to joint constraints or obstacles. The knowledge is present without the need to simulate; it's explicit. Exactly this kind of information will be encoded when successfully learning switching dynamics. Our goal in this work will therefore entail the search for richer representations in the form of latent state space models which encode knowledge about the underlying system dynamics. In turn, we expect this to improve the accuracy of our simulation as well. Such a representation alone could then be used in a reinforcement learning approach that possibly only takes advantage of the learned latent features but not necessarily its learned dynamics.

To learn richer representations, we identify one common problem with prevalent recurrent Variational Autoencoder models (Karl et al., 2017a; Krishnan et al., 2015; Chung et al., 2015; Fraccaro et al., 2016): the non-probabilistic treatment of the transition dynamics often modeled by a powerful nonlinear function approximator. From the history of the Autoencoder to the Variational Autoencoder, we know that in order to detect features in an unsupervised manner, probabilistic treatment of the latent space is paramount. As our starting point, we will build on previously proposed approaches by Krishnan et al. (2017) and Karl et al. (2017a). The latter already made use of locally linear dynamics, but only in a deterministic fashion. We extend their approaches by a stochastic switching LDS model and show that such treatment is vital for learning richer representations and simulation accuracy.

## 2 Background

We consider discretized time-series data consisting of continuous observations $x_t \in \mathcal{X} \subset \mathbb{R}^{n_x}$ and control inputs $u_t \in \mathcal{U} \subset \mathbb{R}^{n_u}$ that we would like to model by corresponding latent states $z_t \in \mathcal{Z} \subset \mathbb{R}^{n_z}$. We'll denote sequences of variables by $x_{1:T} = (x_1, x_2, ..., x_T)$.

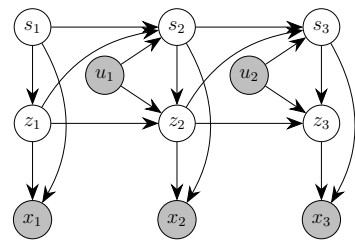

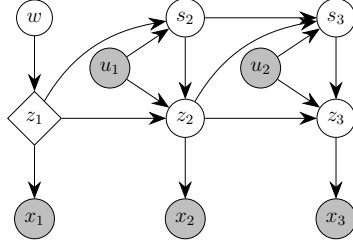

(a) SLDS graphical model.        (b) Our generative model.

Figure 1: (a) $s_t$ denote discrete switch variables, $z_t$ are continuous latent variables, $x_t$ continuous observed variables, $u_t$ are (optional) continuous control inputs. (b) By introducing a special latent variable $w$ used for initial state inference, we want to make explicit that the first step is treated differently from the rest of the sequence.

## 2.1 Switching linear dynamical systems

Switching Linear Dynamical System models (SLDS) enable us to model nonlinear time series data by splitting it into sequences of linear dynamical models. At each time $t = 1, 2, ..., T$, a discrete switch variable $s_t \in 1, ..., M$ chooses of a set LDSs a system which is to be used to transform our continuous latent state $z_t$ to the next time step (Barber, 2012).

$$
\begin{aligned}
z_t &= A(s_t)z_{t-1} + B(s_t)u_{t-1} + \epsilon(s_t) & \epsilon(s_t) &\sim \mathcal{N}(0, Q(s_t)) \\
x_t &= H(s_t)z_t + \eta(s_t) & \eta(s_t) &\sim \mathcal{N}(0, R(s_t))
\end{aligned}
\tag{1}
$$

Here $A \in \mathbb{R}^{n_z \times n_z}$ is the state matrix, $B \in \mathbb{R}^{n_z \times n_u}$ control matrix, $\epsilon$ the transition noise with covariance matrix $Q$ and $\eta$ the emission/sensor noise with covariance matrix $R$. Finally, the observation matrix $H \in \mathbb{R}^{n_x \times n_z}$ defines a linear mapping from latent to observation space which we will replace by a nonlinear transformation parameterized by a neural net. These equations imply the following joint distribution:

$$
p(x_{1:T}, z_{1:T}, s_{1:T} \mid u_{1:T}) = \prod_{t=1}^{T} p(x_t \mid z_t)\, p(z_t \mid z_{t-1}, u_{t-1}, s_t)\, p(s_t \mid z_{t-1}, u_{t-1}, s_{t-1})
\tag{2}
$$

with $p(z_1 \mid z_0, u_0, s_1) = p(z_1)$ being the initial state distribution. The corresponding graphical model is shown in figure 1a.

## 2.2 Stochastic gradient variational Bayes

$$
p(x) = \int p(x, z)\, \mathrm{d}z = \int p(x \mid z)p(z)\, \mathrm{d}z
\tag{3}
$$

Given the simple graphical model in equation (3), Kingma & Welling (2014) and Rezende et al. (2014) introduced the *Variational Autoencoder* (VAE) which overcomes the intractability of posterior inference of $q(z \mid x)$ by maximizing the evidence lower bound (ELBO) of the model log-likelihood.

$$
\mathcal{L}_{\mathrm{ELBO}}(x; \theta, \phi) = \mathbb{E}_{q_\phi(z|x)}[\ln p_\theta(x \mid z)] - D_{\mathrm{KL}}(q_\phi(z \mid x) \,\|\, p(z)) \leq \log p(x)
\tag{4}
$$

Their main innovation was to approximate the intractable posterior distribution by a *recognition network* $q_\phi(z|x)$ from which they can sample via the *reparameterization trick* to allow for stochastic backpropagation through both the recognition and generative model at once. Assuming that the latent state is normally distributed, a simple transformation allows us to obtain a Monte Carlo gradient estimate of $\mathbb{E}_{q_\phi(z|x)}[\ln p_\theta(x|z)]$ w.r.t. to $\phi$. Given that $z \sim \mathcal{N}(\mu, \sigma^2)$, we can generate samples by drawing from an auxiliary variable $\epsilon \sim \mathcal{N}(0, 1)$ and applying the deterministic and differentiable transformation $z = \mu + \sigma\epsilon$.

## 2.3 The concrete distribution

One simple and efficient way to obtain samples $d$ from a $k$-dimensional categorical distribution with class probabilities $\alpha$ is the Gumbel-Max trick:

$$
d = \text{one\_hot}\left(\text{argmax}[g_i + \log \alpha_i]\right), \quad \text{with } g_1, \ldots, g_k \sim \text{Gumbel}(0, 1)
\tag{5}
$$

However, since the derivative of the argmax is 0 everywhere except at the boundary of state changes, where it is undefined, we can't learn a parameterization by backpropagation. The Gumbel-Softmax trick approximates the argmax by a softmax which gives us a probability vector (Maddison et al., 2017; Jang et al., 2017). We can then draw samples via

$$d_k = \frac{\exp((\log \alpha_k + g_k)/\lambda)}{\sum_{i=1}^{n} \exp((\log \alpha_i + g_i)/\lambda)}, \quad \text{with } g_1, \ldots, g_k \sim \text{Gumbel}(0, 1) \tag{6}$$

This softmax computation approaches the discrete argmax as temperature $\lambda \to 0$, for $\lambda \to \infty$ it approaches a uniform distribution.

## 3 RELATED WORK

Our model can be viewed as a Deep Kalman Filter (Krishnan et al., 2015) with structured inference (Krishnan et al., 2017). In our case, structured inference entails another stochastic variable model with parameter sharing inspired by Karl et al. (2017b) and Karl et al. (2017a) which pointed out the importance of backpropagating the reconstruction error through the transition. We are different to a number of stochastic sequential models like Bayer & Osendorfer (2014); Chung et al. (2015); Shabanian et al. (2017); Goyal et al. (2017) by directly transitioning the stochastic latent variable over time instead of having an RNN augmented by stochastic inputs. Fraccaro et al. (2016) has a transition over both a deterministic and a stochastic latent state sequence, wanting to combine the best of both worlds.

Previous models (Watter et al., 2015; Karl et al., 2017a; Fraccaro et al., 2017) have already combined locally linear models with recurrent Variational Autoencoders, however they provide a weaker structural incentive for learning latent variables determining the transition function. Van Steenkiste et al. (2018) approach a similar multi bouncing ball problem (see section 5.1) by first distributing the representation of different balls into their own entities without supervision and then structurally hardwiring a transition function with interactions based on an attention mechanism.

Recurrent switching linear dynamical systems (Linderman et al., 2016) uses message passing for approximate inference, but has restricted itself to low-dimensional observations and a multi-stage training process. Johnson et al. (2016) propose a similar model to ours but combine message passing for discrete switching variables with a neural network encoder for observations learned by stochastic backpropagation. Tackling the problem of propagating state uncertainty over time, various combinations of neural networks for inference and Gaussian processes for transition dynamics have been proposed (Eleftheriadis et al., 2017; Doerr et al., 2018). However, these models have not been demonstrated to work with high-dimensional observation spaces like images. One feature a switching LDS model may learn are interactions which have recently been approached by employing Graph Neural Networks (Battaglia et al., 2016; Kipf et al., 2018). These methods are similar in that they predict edges which encode interactions between components of the state space (nodes).

## 4 PROPOSED APPROACH

Our goal is to fit a series of continuous state $z_{1:T}$ and switching variables $s_{2:T}$ to a given sequence of observations $x_{1:T}$. We assume a nonlinear mapping between observations and latent space which we generally approximate by neural networks, apart from the transition which is modeled by a locally linear function. Our generative model is shown in figure 1b an our inference model in figure 2a.

### 4.1 GENERATIVE MODEL

Our generative model for a single $x_t$ is described by

$$p(x_t) = \int_{s_{\leq t}} \int_{z_{\leq t}} p(x_t \mid z_t) p(z_t \mid z_{t-1}, s_t, u_{t-1}) p(s_t \mid s_{t-1}, z_{t-1}, u_{t-1}) p(z_{t-1}, s_{t-1}) \tag{7}$$

which is close to the one of the original SLDS model (see figure 1a). Latent states $z_t$ are continuous and represent the state of the system while states $s_t$ are the switching variables determining the transition. We approximate the discrete switching variables by a continuous relaxation, namely

the Concrete distribution.[1] Differently to the original model, we do not condition the likelihood of the current observation $p_\theta(x_t \mid z_t)$ directly on the switching variables. This limits the influence of the switching variables to choosing a proper transition dynamic for the continuous latent space. The likelihood model is parameterized by a neural network with either a Gaussian or a Bernoulli distribution as output depending on the data.

There is both a transition on the continuous states $z_t$ and discrete latent states $s_t$. For the continuous state transition $p(z_t \mid z_{t-1}, s_t, u_{t-1})$ we follow (1) and maintain a set of $M$ base matrices $\{(A^{(i)}, B^{(i)}, Q^{(i)}) \mid \forall i.\ 0 < i < M\}$ as our linear dynamical systems to choose from. For the transition on discrete latent states $p(s_t \mid s_{t-1}, z_{t-1}, u_{t-1})$, we usually require the learning of a Markov transition matrix. However, since we approximate our discrete switching variables by a continuous relaxation, we can parameterize this transition by a neural network. Therefore, our entire generative model can be learned end-to-end by (stochastic) backpropagation. Finally, the resulting dynamics matrices are computed through a linear combination of the base matrices:

$$A_t(s_t) = \sum_{i=1}^{M} s_t^{(i)} A^{(i)}, \qquad B(s_t) = \sum_{i=1}^{M} s_t^{(i)} B^{(i)}, \qquad Q(s_t) = \sum_{i=1}^{M} s_t^{(i)} Q^{(i)} \qquad (8)$$

Both transition models – the continuous state transition $p_\theta(z_t \mid z_{t-1}, s_t, u_{t-1})$ and concrete switching variables transition $p_\theta(s_t \mid s_{t-1}, z_{t-1}, u_{t-1})$ – are shared with the inference model which is key for good performance.

$$p_\theta(z_t \mid z_{t-1}, s_t, u_{t-1}) = \mathcal{N}(\mu, \sigma^2) \qquad \text{where}\ \ [\mu, \sigma^2] = f_\theta(z_{t-1}, s_t, u_{t-1})$$
$$p_\theta(s_t \mid s_{t-1}, z_{t-1}, u_{t-1}) = \text{Concrete}(\alpha, \lambda_{\text{prior}}) \qquad \text{where}\ \ \alpha = g_\theta(z_{t-1}, s_{t-1}, u_{t-1}) \qquad (9)$$

## 4.2 INFERENCE

### 4.2.1 STRUCTURED INFERENCE OF CONTINUOUS LATENT STATE

We split our inference model $q_\phi(z_t \mid z_{t-1}, s_t, x_{\geq t}, u_{\geq t-1})$ into two parts: 1) transition model $q_{\text{trans}}(z_t \mid z_{t-1}, s_t, u_{t-1})$ and 2) inverse measurement model $q_{\text{meas}}(z_t \mid x_{\geq t}, u_{\geq t})$ as previously proposed in Karl et al. (2017b). This split allows us to reuse our generative transition model in place of $q_{\text{trans}}(z_t \mid z_{t-1}, s_t, u_{t-1})$. This sharing of variables is essential for good performance as it forces the reconstruction error to be backpropagated through the transition model. For practical reasons, we only share the computation of the transition mean $\mu_{\text{trans}}$ but not the variance $\sigma^2_{\text{trans}}$ between inference and generative model. Both parts, $q_{\text{meas}}$ and $q_{\text{trans}}$, will give us independent predictions about the new state $z_t$ which will be combined in a manner akin to a Bayesian update in a Kalman Filter.

$$q_\phi(z_t \mid z_{t-1}, s_t, x_{\geq t}, u_{\geq t-1}) \propto q_{\text{meas}}(z_t \mid x_{\geq t}, u_{\geq t}) \times q_{\text{trans}}(z_t \mid z_{t-1}, s_t, u_{t-1}) = \mathcal{N}(\mu_q, \sigma_q^2)$$
$$q_{\text{meas}}(z_t \mid x_{\geq t}, u_{\geq t}) = \mathcal{N}(\mu_{\text{meas}}, \sigma^2_{\text{meas}})\ \text{where}\ [\mu_{\text{meas}}, \sigma^2_{\text{meas}}] = h_\phi(x_{\geq t}, u_{\geq t}) \qquad (10)$$
$$q_{\text{trans}}(z_t \mid z_{t-1}, s_t, u_{t-1}) = \mathcal{N}(\mu_{\text{trans}}, \sigma^2_{\text{trans}})\ \text{where}\ [\mu_{\text{trans}}, \sigma^2_{\text{trans}}] = f_\theta(z_{t-1}, s_t, u_{t-1})$$

The densities of $q_{\text{meas}}$ and $q_{\text{trans}}$ are multiplied resulting in another Gaussian density:

$$\mu_q = \frac{\mu_{\text{trans}}\sigma^2_{\text{meas}} + \mu_{\text{meas}}\sigma^2_{\text{trans}}}{\sigma^2_{\text{meas}} + \sigma^2_{\text{trans}}}, \qquad \sigma_q^2 = \frac{\sigma^2_{\text{meas}}\sigma^2_{\text{trans}}}{\sigma^2_{\text{meas}} + \sigma^2_{\text{trans}}} \qquad (11)$$

This update scheme is highlighted in figure 2b.

We found empirically that conditioning the inverse measurement model $q_{\text{meas}}(z_t \mid x_{\geq t}, u_{\geq t})$ solely on the current observation $x_t$ instead of the entire remaining trajectory to lead to better results. We hypothesize that the recurrent model needlessly introduces very high-dimensional and complicated dynamics which are harder to approximate with our locally linear transition model.

For the initial state $z_1$ we do not have a conditional prior from the transition model as in the rest of the sequence. Other methods (Krishnan et al., 2015) have used a standard normal prior, however this is not a good fit. We therefore decided that instead of predicting $z_1$ directly to predict an auxiliary

---

[1]As an ablation study, we will compare this to modeling switching variables by a Gaussian distribution.

variable $w$ that is then mapped deterministically to a starting state $z_1$. A standard Gaussian prior is then applied to $w$. Alternatively, we could specify a more complex or learned prior for the initial state like the VampPrior (Tomczak & Welling, 2017). Empirically, this has lead to worse results.

$$q_\phi(w \mid x_{1:T}, u_{1:T}) = \mathcal{N}\big(w; \mu_w, \sigma_w^2\big) \quad \text{where} \quad [\mu_w, \sigma_w^2] = i_\phi(x_{1:T}, u_{1:T})$$
$$z_1 = f_\phi(w) \tag{12}$$

While we could condition on the entire sequence, we restrict it to just the first couple of observations.

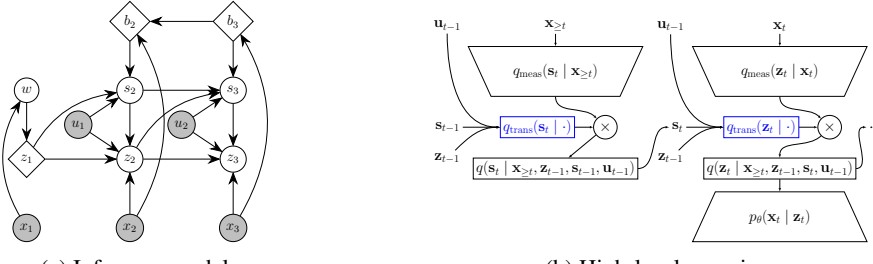

(a) Inference model.  (b) High-level overview.

Figure 2: (a) Depicts the inference model. $b_t$ is the hidden state of the backward RNN of $q_\phi(s_t \mid x_{\geq t}, u_{\geq t})$. Initial inference of $w$ may be conditioned on the entire sequence of observations, or just a subsequence. We've omitted the arrows for sake of clarity for the rest of the graph. (b) Shows schematically how we combine the transition with the inverse measurement model in the inference network. Transitions (in blue) are (partially) shared with the generative model.

### 4.2.2 INFERENCE OF SWITCHING VARIABLES

Following Maddison et al. (2017) and Jang et al. (2017), we can reparameterize a discrete latent variable with the Gumbel-softmax trick. Again, we split our inference network $q_\phi(s_t \mid s_{t-1}, z_{t-1}, x_{\geq t}, u_{\geq t-1})$ in an identical fashion into two components: 1) Transition model $q_{\text{trans}}(s_t \mid s_{t-1}, z_{t-1}, u_{t-1})$ and 2) inverse measurement model $q_{\text{meas}}(s_t \mid x_{\geq t}, u_{\geq t})$. The transition model is again shared with the generative model and is implemented via a neural network as we potentially require quick changes to chosen dynamics. The inverse measurement model is parametrized by a backward LSTM. However, for the case of concrete variables, we cannot do the same Gauss multiplication as in the previous case. Therefore, we let each network predict the logits of a Concrete distribution and our inverse measurement model $q_\phi(s_t \mid x_{\geq t}, u_{\geq t})$ produces an additional vector $\gamma$, which determines the value of a gate deciding how the two predictions are to be weighted:

$$q_\phi(s_t \mid s_{t-1}, z_{t-1}, x_{\geq t}, u_{\geq t-1}) = \text{Concrete}(\alpha, \lambda_{\text{posterior}}) \quad \text{with} \quad \alpha = \gamma \alpha_{\text{trans}} + (1 - \gamma)\alpha_{\text{meas}}$$
$$q_{\text{meas}}(s_t \mid x_{\geq t}, u_{\geq t}) = \text{Concrete}(\alpha_{\text{meas}}, \lambda_{\text{posterior}}) \quad \text{where} \quad [\alpha_{\text{meas}}, \gamma] = k_\phi(x_{\geq t}, u_{\geq t}) \tag{13}$$
$$q_{\text{trans}}(s_t \mid s_{t-1}, z_{t-1}, u_{t-1}) = \text{Concrete}(\alpha_{\text{trans}}, \lambda_{\text{prior}}) \quad \text{where} \quad \alpha = g_\theta(z_{t-1}, s_{t-1}, u_{t-1})$$

The temperatures $\lambda_{\text{posterior}}$ and $\lambda_{\text{prior}}$ are set as a hyperparameter and can be set differently for the prior and approximate posterior. The gating mechanism gives the model the option to balance between prior and approximate posterior. If the prior is good enough to explain the next observation, $\gamma$ will be pushed to 1 which ignores the measurement and minimizes the KL between prior and posterior by only propagating the prior. If the prior is not sufficient, information from the inverse measurement model can flow by decreasing $\gamma$ and incurring a KL penalty.

Since the concrete distribution is a relaxation of the categorical, our sample will not be a one-hot vector, but a vector whose elements sum up to 1. We face two options here: we could take a categorical sample by choosing the linear system corresponding to the highest value in the sample (hard forward pass) and only use the relaxation for our backward pass. This, however, means that we will follow a biased gradient. Alternatively, we can use the relaxed version for our forward pass and aggregate the linear systems based on their corresponding weighting (see (8)). Here, we lose the discrete switching of linear systems, but maintain a valid lower bound. We note that the hard forward pass has led to worse results and focus on the soft forward pass for this paper.

Lastly, we could go further away from the theory and instead treat the switching variables also as normally distributed. If this worked better than the approach with Concrete variables, it would

highlight still existing optimization problems of discrete random variables. As such, it will act as an ablation study for our model. The mixing coefficients for linear systems would then be determined by a linear combination of these latent variables:

$$\alpha = \text{softmax}(Ws_t + b) \in \mathbb{R}^M \tag{14}$$

Our inference scheme for normally distributed switching variables is then identical to the one described in the previous section. We compare both approaches throughout our experimental section.

### 4.3 TRAINING

Our objective function is the commonly used evidence lower bound for our hierarchical model.

$$
\begin{aligned}
\mathcal{L}_{\theta,\phi}(x_{1:T} \mid u_{1:T}) \geq \quad & \mathbb{E}_{q_\phi(z_{1:T}, s_{1:T} \mid x_{1:T})}[\log p_\theta(x_{1:T} \mid z_{1:T}, s_{1:T}, u_{1:T})] \\
& - D_{\text{KL}}(q_\phi(z_{1:T}, s_{1:T} \mid x_{1:T}, u_{1:T}) \mid\mid p(z_{1:T}, s_{1:T} \mid u_{1:T}))
\end{aligned} \tag{15}
$$

We choose to factorize over time, so the loss for a single observation $x_t$ becomes:

$$
\begin{aligned}
\mathcal{L}_{\theta,\phi}(x_t \mid u_{1:T}) = & \mathbb{E}_{q_\phi\left(s_t \mid s_{t-1}, z_{t-1}, x_{\geq t}, u_{\geq t-1}\right)}\Big[\mathbb{E}_{q_\phi\left(z_t \mid s_t, z_{t-1}, x_{\geq t}, u_{\geq t-1}\right)}[\log p_\theta(x_t \mid z_t)]\Big] \\
& - \mathbb{E}_{s_{t-1}}\Big[\mathbb{E}_{z_{t-1}}[D_{\text{KL}}(q_\phi(s_t \mid s_{t-1}, z_{t-1}, x_{\geq t}, u_{\geq t-1}) \mid\mid p_\theta(s_t \mid s_{t-1}, z_{t-1}, u_{t-1}))]\Big] \\
& - \mathbb{E}_{z_{t-1}}\Big[\mathbb{E}_{s_t}[D_{\text{KL}}(q_\phi(z_t \mid z_{t-1}, s_t, x_{\geq t}, u_{\geq t-1}) \mid\mid p_\theta(z_t \mid z_{t-1}, s_t, u_{t-1}))]\Big]
\end{aligned} \tag{16}
$$

The full derivation can be found in appendix A. We learn the parameters of our model by backpropagation through time and we (generally) approximate the expectations with one sample by using the reparametrization trick. The exception is the KL between two Concrete random variables in which case we take 10 samples for the approximation. For the KL on the switching variables, we further introduce a scaling factor $\beta < 1$ (as first suggested in Higgins et al. (2016), although they suggested increasing the KL term) to down weigh its importance. More details on the training procedure can be found in appendix B.2.

## 5 EXPERIMENTS

In this section, we evaluate our approach on a diverse set of physics and robotics simulations based on partially observable system states or high-dimensional images as observations. We show that our model outperforms previous models and that our switching variables learn meaningful representations.

Models we compare to are Deep Variational Bayes Filter (DVBF) (Karl et al., 2017a), DVBF Fusion (Karl et al., 2017b) (called fusion as they do the same Gauss multiplication in the inference network) which is closest to our model but doesn't have a stochastic treatment of the transition, the Kalman VAE (KVAE) (Fraccaro et al., 2017) and a LSTM (Hochreiter & Schmidhuber, 1997).

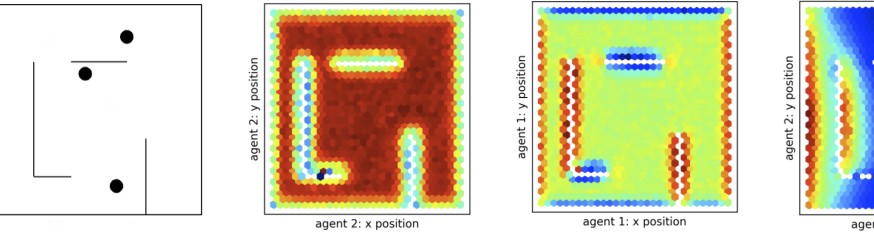

(a) Multi agent maze environment.   (b) Variable encoding free space for agent 2.   (c) Variable encoding walls for agent 1.   (d) System activation for deterministic transition.

Figure 3: Figures (b) and (c) depict an agent's position colored by the average value of a single latent variable $s$ marginalized over all control inputs $u$ and velocities. Figure (d) highlights a representative activation for a single transition system for the deterministic treatment of the transition dynamics. It doesn't generalize to the entire maze and stays fairly active in proximity to the wall.

## 5.1 MULTIPLE BOUNCING BALLS IN A MAZE

Our first experiment is a custom 3-agent maze environment simulated with Box2D. Each agent is fully described by its $x$ and $y$ coordinates and its current velocity and has the capability to accelerate in either direction. We learn in a partially observable setting and limit the observations to the agents' positions, therefore $x \in \mathbb{R}^6$ while the true state space is in $\mathbb{R}^{12}$ and $u \in \mathbb{R}^6$. First, we train a linear regression model on the latent space $z$ to see if we have recovered a linear encoding of the unobserved velocities. We achieve an R2 score of 0.92 averaged over all agents and velocity directions.

Our focus shifts now to our switching variables which we expect to encode interactions with walls. We provide a visual confirmation of that in figure 3 where we see switching variables encoding all space where there is no interaction in the next time step, and variables which encode walls, distinguishing between vertical and horizontal ones. In figure 3d one can see show that if the choice of locally linear transition is treated deterministically, we don't learn global features of the same kind. To confirm our visual inspection, we train a simple decision tree based on latent space $s$ in order to predict interaction with a wall. Here, we achieve an F1 score of $0.46$. It is difficult to say what a good value should look like as collisions with low velocity are virtually indistinguishable from no collision.

We compare our prediction quality to several other methods in table 1 where we outperform all of our chosen baselines. Also, modeling switching variables by a Normal distribution outperforms the Concrete distribution in all of our experiments. Aside from known practical issues with training a discrete variable via backpropagation, we explore one reason why that may be in section 5.4, which is the greater susceptibility to the scale of temporal discretization. We provide plots of predicted trajectories in appendix D. Transitioning multiple agents with a single transition matrix comes with scalability issues with regards to switching dynamics which we explore further in appendix C.

Table 1: Mean squared error (MSE) on predicting future observations. Static refers to constantly predicting the first observation of the sequence.

|  | REACHER | | | 3-BALL MAZE | | |
| --- | --- | --- | --- | --- | --- | --- |
| PREDICTION STEPS | 1 | 5 | 10 | 1 | 5 | 10 |
| STATIC | 5.80E-02 | 5.36E-01 | 1.25E+00 | 1.40E-02 | 5.74E-01 | 2.65E+00 |
| LSTM | 3.07E-01 | 7.76E-01 | 1.22E+00 | 7.20E-02 | 1.58E-01 | 2.60E-01 |
| DVBF | 1.10E-01 | 3.08E-01 | 6.07E-01 | 6.20E-02 | 1.36E-01 | 1.82E-01 |
| DVBF FUSION | 4.90E-03 | 2.97E-02 | 8.25E-02 | 4.33E-03 | 2.03E-02 | 4.88E-02 |
| OURS (CONCRETE) | 1.06E-02 | 5.73E-02 | 1.56E-01 | 2.28E-03 | 1.22E-02 | 3.40E-02 |
| OURS (NORMAL) | **3.39E-03** | **1.85E-02** | **4.97E-02** | **1.30E-03** | **5.52E-03** | **1.38E-02** |

## 5.2 REACHER

We then evaluate our model on the Roboschool reacher environment. To make things more interesting, we learn only on partial observations, removing time derivative information (velocities), leaving us with just the positions or angles of various joints as observations. Table 1 shows a comparison of various methods on predicting the next couple of time steps. One critical point is the possible collision[2] between lower and upper joint which is one we'd like our model to capture. We again learn a linear classifier based on latent space $s$ to see if this is successfully encoded and reach an F1 score of $0.46$.

## 5.3 BALL IN A BOX ON IMAGE DATA

Finally, we evaluate our method on high-dimensional image observations using the single bouncing ball environment used by Fraccaro et al. (2017). They simulated 5000 sequences of 20 time steps each of a ball moving in a two-dimensional box, where each video frame is a $32 \times 32$ binary image. There are no forces applied to the ball, except for the fully elastic collisions with the walls. Initial position and velocity are randomly sampled.

---

[2]We roughly identify a collision to be the point where the lower joint decelerates by over a fixed value of 2.

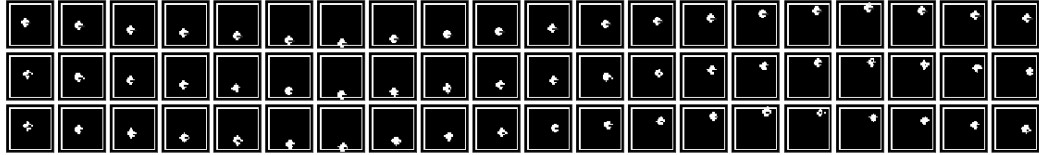

Figure 4: First row: data, second row: filtered reconstructions, third row: predictions. The first 4 steps are used to find a stable starting state, predictions start with step 5.

In figure 5a we compare our model to both the smoothed and generative version of the KVAE. The smoothed version receives the final state of the trajectory after the $n$ predicted steps which is fed into the smoothing capability of the KVAE. One can see that our model learns a better transition model, even outperforming the smoothed KVAE for longer sequences. For short sequences, KVAE performs better which highlights the value of it disentangling the latent space into separate object and dynamics representation. A sample trajectory is plotted in figure 4.

## 5.4 SUSCEPTIBILITY TO THE SCALE OF TEMPORAL DISCRETIZATION

In this section, we'd like to explore how the choice of $\Delta t$ when discretizing a system influences our results. In particular, we'd expect our model with discrete (concrete) switching latent variables to be more susceptible to it than when modeled by a continuous distribution. This is because in the latter case the switching variables can scale the various matrices more freely, while in the former scaling up one system necessitates scaling down another. For empirical comparison, we go back to our custom maze environment (this time with only one agent as this is not pertinent to our question at hand) and learn the dynamics on various discretization scales. Then we compare the absolute error's growth for both approaches in figure 5b which supports our hypothesis. While the discrete approximation even outperforms for small $\Delta t$, there is a point where it rapidly becomes worse and gets overtaken by the continuous approximation. This suggests that $\Delta t$ was simply chosen to be too large in both the reacher and the ball in a box with image observations experiment.

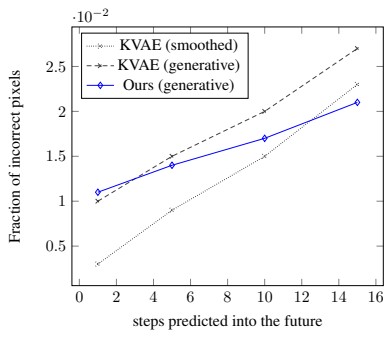

(a) Fraction of incorrectly predicted pixels.

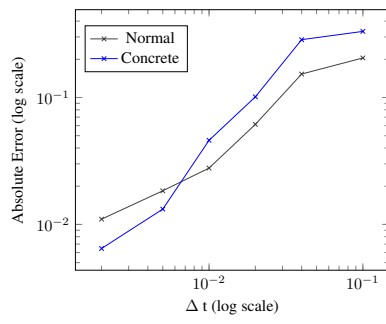

(b) Discretization scale susceptibility.

Figure 5: (a) Our dynamics model is outperforming even the smoothed KVAE for longer trajectories. (b) Modeling switching variables as Concrete random variables scales less favorably.

## 6 DISCUSSION

We want to emphasize some subtle differences to previously proposed architectures that make an empirical difference, in particular for the case when $s_t$ is chosen to be continuous. In Watter et al. (2015) and Karl et al. (2017a), the latent space is already used to draw transition matrices, however they do not extract features such as walls or joint constraints. There are a few key differences from our approach. First, our latent switching variables $s_t$ are only involved in predicting the current observation $x_t$ through the transition selection process. The likelihood model therefore doesn't need to learn to ignore some input dimensions which are only helpful for reconstructing future observations

but not the current one. There is also a clearer restriction on how $s_t$ and $z_t$ may interact: $s_t$ may now only influence $z_t$ by determining the dynamics, while previously $z_t$ influenced both the choice of transition function as well as acted inside the transition. These two opposing roles lead to conflicting gradients as to what should be improved. Furthermore, the learning signal for $s_t$ is rather weak so that scaling down the KL-regularization was necessary to detect good features. Lastly, a (locally) linear transition may not be a good fit for variables determining dynamics as such variables may change very abruptly.

## 7 CONCLUSION

We have shown that our construction of using switching variables encourages learning a richer and more interpretable latent space. In turn, the richer representation led to an improvement of simulation accuracy in various tasks. In the future, we'd like to look at other ways to approximate the discrete switching variables and exploit this approach for model-based control on real hardware systems. Furthermore, addressing the open problem of disentangling latent spaces is essential to fitting simple dynamics and would lead to significant improvements of this approach.

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

## A   LOWER BOUND DERIVATION

For brevity we omit conditioning on control inputs $u_{1:T}$.

$$
\begin{aligned}
\log p(x_T) &= \log \int_{z_{1:T}} \int_{s_{1:T}} q_\phi(s_{1:T}, z_{1:T} \mid x_{1:T}) \frac{p_\theta(x_{1:T} \mid z_{1:T}) p_\theta(z_{1:T}, s_{1:T})}{q_\phi(s_{1:T}, z_{1:T} \mid x_{1:T})} \\
&\geq \int_{z_{1:T}} \int_{s_{1:T}} q_\phi(s_{1:T}, z_{1:T} \mid x_{1:T}) \log \frac{p_\theta(x_{1:T} \mid z_{1:T}) p_\theta(z_{1:T}, s_{1:T})}{q_\phi(s_{1:T}, z_{1:T} \mid x_{1:T})} \\
&= \sum_{t=1}^{T} \mathbb{E}_{s_t} [\mathbb{E}_{z_t} [p(x_t \mid z_t, s_t)]] - D_{\mathrm{KL}}(q(z_{1:T}, s_{1:T} \mid x_{1:T}) \,\|\, p(z_{1:T}, s_{1:T}))
\end{aligned}
$$

### A.1   FACTORIZATION OF THE KL DIVERGENCE

The dependencies on data $x_T$ and $u_T$ as well as parameters $\phi$ and $\theta$ are omitted in the following for convenience.

$$
D_{\mathrm{KL}}(q(z_1, s_2, \ldots, s_T, z_T) \,\|\, p(z_1, s_2, \ldots, s_T, z_T))
$$

*(Factorization of the variational approximation)*

$$
= \int_{z_1} \int_{s_2} \cdots \int_{s_T} \int_{z_T} q(z_1) q(s_2 \mid z_1) \ldots q(s_T \mid z_{T-1}, s_{T-1}) q(z_T \mid z_{T-1}, s_T)
$$
$$
\log \frac{q(z_1) q(s_2 \mid z_1) \ldots q(s_T \mid z_{T-1}, s_{T-1}) q(z_T \mid z_{T-1}, s_T)}{p(z_1, s_2, \ldots, s_T, z_T)}
$$

*(Factorization of the prior)*

$$
= \int_{z_1} \int_{s_2} \cdots \int_{s_T} \int_{z_T} q(z_1) q(s_2 \mid z_1) \ldots q(s_T \mid z_{T-1}, s_{T-1}) q(z_T \mid z_{T-1}, s_T)
$$
$$
\log \frac{q(z_1) q(s_2 \mid z_1) \ldots q(s_T \mid z_{T-1}, s_{T-1}) q(z_T \mid z_{T-1}, s_T)}{p(z_1) p(s_2 \mid z_1) \ldots p(s_T \mid z_{T-1}, s_{T-1}) p(z_T \mid z_{T-1}, s_T)}
$$

*(Expanding the logarithm by the product rule)*

$$
= \int_{z_1} q(z_1) \log \frac{q(z_1)}{p(z_1)} + \int_{z_1} \int_{s_1} q(z_1) q(s_1 \mid z_1) \log \frac{q(s_1 \mid z_1)}{p(s_1 \mid z_1)}
$$
$$
+ \sum_{t=2}^{T} \int_{z_1} \int_{s_2} \cdots \int_{s_T} \int_{z_T} q(z_1) q(s_2 \mid z_1) \ldots q(z_T \mid z_{T-1}, s_T) \log \frac{q(z_t \mid z_{t-1}, s_t)}{p(z_t \mid z_{t-1}, s_t)}
$$
$$
+ \sum_{t=3}^{T} \int_{z_1} \int_{s_2} \cdots \int_{s_T} \int_{z_T} q(z_1) q(s_2 \mid z_1) \ldots q(z_T \mid z_{T-1}, s_T) \log \frac{q(s_t \mid z_{t-1}, s_{t-1})}{p(s_t \mid z_{t-1}, s_{t-1})}
$$

*(Ignoring constants)*

$$
= D_{\mathrm{KL}}(q(z_1) \,\|\, p(z_1)) + \mathbb{E}_{z_1 \sim q(z_1)} [D_{\mathrm{KL}}(q(s_2 \mid z_1) \,\|\, p(s_2 \mid z_1))]
$$
$$
+ \sum_{t=2}^{T-1} \mathbb{E}_{s_t, z_{t-1}} [D_{\mathrm{KL}}(q(z_t \mid z_{t-1}, s_t) \,\|\, p(z_t \mid z_{t-1}, s_t))]
$$
$$
+ \sum_{t=3}^{T-1} \mathbb{E}_{s_{t-1}, z_{t-1}} [D_{\mathrm{KL}}(q(s_t \mid z_{t-1}, s_{t-1}) \,\|\, p(s_t \mid z_{t-1}, s_{t-1}))]
$$

Table 2: Dimensionality of environments.

| Dimensionality of | Observation Space | Control Input Space | Ground Truth State Space |
|---|---|---|---|
| Reacher | 7 | 2 | 9 |
| Hopper | 8 | 3 | 15 |
| Multi Agent Maze | 4 | 6 | 12 |
| Image Ball in Box | $32 \times 32$ | 0 | 4 |

## B    DETAILS OF THE EXPERIMENTAL SETUP

### B.1    ENVIRONMENTS

#### B.1.1    ROBOSCHOOL REACHER

To generate data, we follow a Uniform distribution $\mathcal{U} \sim [-1, 1]$ as the exploration policy. Before we record data, we take 20 warm-up steps in the environment to randomize our starting state. We take the data as is without any other preprocessing.

#### B.1.2    MULTI AGENT MAZE

Observations are normalized to be in $[-1, 1]$. Both position and velocity is randomized for the starting state. We again follow a Uniform distribution $\mathcal{U} \sim [-1, 1]$ as the exploration policy.

### B.2    TRAINING

Overall, training the Concrete distribution has given us the biggest challenge as it was very susceptible to various hyperparameters. We made use of the fact that we can use a different temperature for the prior and approximate posterior (Maddison et al., 2017) and we do independent hyperparameter search over both. For us, the best values were $0.75$ for the posterior and $2$ for the prior. Additionally, we employ an exponential annealing scheme for the temperature hyperparameter of the Concrete distribution. This leads to a more uniform combination of base matrices early in training which has two desirable effects. First, all matrices are scaled to a similar magnitude, making initialization less critical. Second, the model initially tries to fit a globally linear model, leading to a good starting state for optimization. We also tried increasing the number of samples taken (up to $100$) to approximate the KL between the Concrete distributions, however we have not observed an improvement of performance. We therefore restrict ourselves to 10 samples for all experiments.

In all experiments, we train everything end-to-end with the ADAM optimizer.(Kingma & Ba, 2015) We start with learning rate of $5e-4$ and use an exponential decay schedule with rate $0.97$ every $2000$ iterations.

### B.3    NETWORK ARCHITECTURE

For most networks, we use MLPs implemented as residual nets (He et al., 2016) with ReLU activations.

Networks used for the reacher and maze experiments.

- $q_{\text{meas}}(z_t \mid \cdot)$: MLP consisting of two residual blocks with 256 neurons each. We only condition on the current observation $x_t$ although we could condition on the entire sequence. This decision was taken based on empirical results.

- $q_{\text{trans}}(z_t \mid \cdot)$: In the case of Concrete random variables, we just combine the base matrices and apply the transition dynamics to $z_{t-1}$. For the Normal case, the combination of matrices is preceded by a linear combination with softmax activation. (see equation 14)

- $q_{\text{meas}}(s_t \mid \cdot)$: is implemented by a backward LSTM with 256 hidden units. We reuse the preprocessing of $q_{meas}(z_t \mid x_t)$ and take the last hidden layer of that network as the input to the LSTM.

- $q_{\text{trans}}(s_t \mid \cdot)$: MLP consisting of one residual block with 256 neurons.
- $q_{\text{initial}}(w \mid \cdot)$: MLP consisting of two residual block with 256 neurons optionally followed by a backward LSTM. We only condition on the first 3 or 4 observations for our experiments.
- $q_{\text{initial}}(s_2)$: The first switching variable in the sequence has no predecessor. We therefore require a replacement for $q_{trans}(s_t \mid \cdot)$ in the first time step, which we achieve by independently parameterizing another MLP.
- $p(x_t \mid z_t)$: MLP consisting of two residual block with 256 neurons.
- $p(z_t \mid \cdot)$: Shared parameters with $q_{trans}(z_t \mid \cdot)$.
- $p(s_t \mid \cdot)$: Shared parameters with $q_{trans}(s_t \mid \cdot)$.

We use the same architecture for the image ball in a box experiment, however we increase number of neurons of $q_{\text{meas}}(z_t \mid \cdot)$ to 1024.

## B.4 HYPERPARAMETERS

Table 3: Overview of hyperparameters.

|  | Multi Agent Maze | Reacher | Image Ball in Box |
|---|---|---|---|
| # episodes | 50000 | 20000 | 5000 |
| episode length | 20 | 30 | 20 |
| batch size | 256 | 128 | 256 |
| dimension of $z$ | 32 | 16 | 8 |
| dimension of $s$ | 16 | 8 | 8 |
| posterior temperature | 0.75 | 0.75 | 0.67 |
| prior temperature | 2 | 2 | 2 |
| temperature annealing steps | 100 | 100 | 100 |
| temperature annealing rate | 0.97 | 0.97 | 0.98 |
| $\beta$ (KL-scaling of switching variables) | 0.1 | 0.1 | 0.1 |

## C ON SCALING ISSUES OF SWITCHING LINEAR DYNAMICAL SYSTEMS

Let's consider a simple representation of a ball in a rectangular box where its state is represented by its position and velocity. Given a small enough $\Delta t$, we can approximate the dynamics decently by just 3 systems: no interaction with the wall, interaction with a vertical or horizontal wall (ignoring the corner case of interacting with two walls at the same time). Now consider the growth of required base systems if we increase the number of balls in the box (even if these balls cannot interact with each other). We would require a system for all combinations of a single ball's possible states: $3^2$. This will grow exponentially with the number of balls in the environment.

One way to alleviate this problem that requires only a linear growth in base systems is to independently turn individual systems on and off and let the resulting system the sum of all activated systems. A base system may then represent solely the transition for a single ball being in specific state, while the complete system is then a combination of $N$ such systems where $N$ is the number of balls. Practically, this can be achieved by replacing the softmax by a sigmoid activation function or by replacing the categorical variable $s$ of dimension $M$ by $M$ Bernoulli variables indicating whether a single system is active or not. We do this for our multiple agents in a maze environment.

Theoretically, a preferred approach would be to disentangle multiple systems (like balls, joints) and apply transitions only to their respective states. This, however, would require a proper and unsupervised separation of (mostly) independent components. We defer this to future work.

## D FURTHER RESULTS

### D.1 3-AGENT MAZE

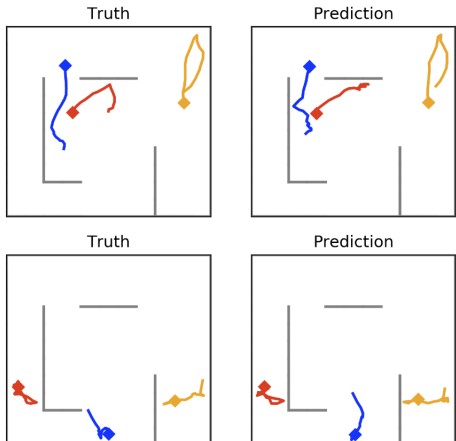

Figure 6: Comparison of actual and predicted 20 step trajectories. The diamond marker denotes the starting position of a trajectory.

### D.2 IMAGE BALL IN A BOX

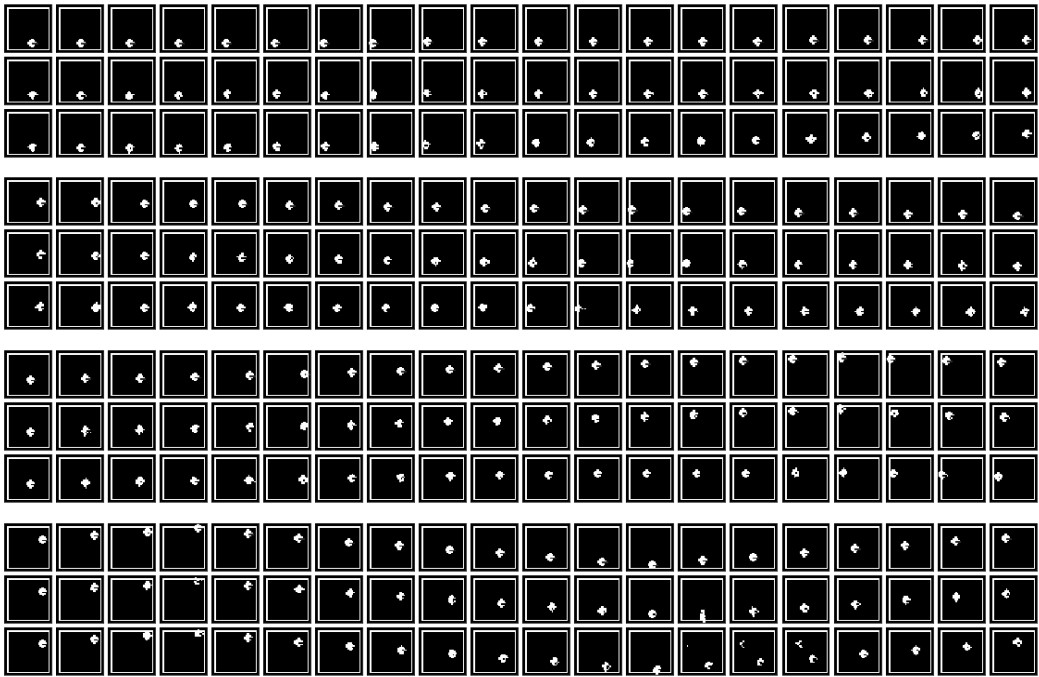

Figure 7: First row: data, second row: reconstructions, third row: predictions. The first 4 steps are used to find a stable starting state, predictions start with step 5.

