# OpenReview forum: "Switching Linear Dynamics for Variational Bayes Filtering"
_ICLR.cc/2019/Conference_

### Official Review · AnonReviewer1 · 2018-10-24
**Interesting paper showing how to use switching variables in deep probabilistic temporal models**

**Rating:** 7
**Confidence:** 5

**Review:**

This paper proposes a deep probabilistic model for temporal data that leverages latent variables to switch between different learned linear dynamics. The probability distributions are parameterized by deep neural networks and learning is performed end-to-end with amortized variational inference using inference networks.

There has been a lot of recent research trying to combine probabilistic models and deep learning to define powerful transition models that can be learned in an unsupervised way, to be used for model-based RL. This paper fits in this research area, and presents a nice combination of several interesting ideas presented in related works (switching variables, structured inference networks, merging updates as in the Kalman filter). The novelty of this paper in terms of original ideas is limited, the novel part lies in the clever combination of known approaches.

The paper reads well, but I found the explanation and notation in section 4 quite confusing (although easy to improve). The authors propose a structured variational approximation, but the factorization assumptions are not clear from the notation (I had to rely on Figure 2a to fully understand them).
- In the first line of equation 7 it seems that the variational approximation q_phi for z_t only depends on x_t, but it is actually dependent also on the future x through s_t and q_meas
- The first line of section 4.1.1 shows that q_phi depends on x_{1:T}. However from figure 2a it seems that it only directly depends on x_{t:T}, and that the dependence on x_{1:t-1} is modelled through the dependence on z_{t-1}.
- Is there a missing s_t in q_trans in the first line of (7)?
- why do you keep the dependence on future outputs in q_meas if it is not used in the experiments and not shown in figure 2a? It only makes the notation more confusing.
- You use f_phi to denote all the function in 4.1.1 (with different inputs). It would be clearer to use a different letter or for example add numbers (e.g. f^1_\phi)
- Despite being often done in VAE papers, it feels strange to me to introduce the inference model (4.1) before the generative model (4.2), as the inference model defines an approximation to the true posterior which is derived from the generative model. One could in principle use other type of approximate inference techniques while keeping the generative model unchanged.

It is difficult for me to understand how useful are in practice the switching variables. Reading the first part of the paper it seems that the authors will use discrete random variables, but they actually use for s_t continuous relaxiations of discrete variables (concrete distribution), or gaussian variables. As described in appendix B2 by the authors, training models with such continuous relaxations is often challenging in terms of hyper-parameter tuning. One may even wonder if it is worth the effort: could you have used instead a deterministic s_t parameterized for example as a bidirectional LSTM with softmax output? This may give equivalent results and remove a lot of complexity. Also, the fact that the gaussian switching variables perform better in the experiments is an indication that this may be the case.

To be able to detect walls the z variables basically need to learn to represent the position of the agent and encoding the information on the position of the walls in the connection to s_t.  Would you then need to train the model from scratch for any new environment?

Minor comment:
- in the softmax equation (6) there are missing brackets: lambda is at the denominator both for g and the log

---

> ### Author Response · Authors · 2018-11-19
> **Thanks for the feedback**
>
> - The first line of section 4.1.1 shows that q_phi depends on x{1:T}. However from figure 2a it seems that it only directly depends on x{t:T}, and that the dependence on x{1:t-1} is modelled through the dependence on z{t-1}.
>
> Thanks, this is indeed a bit confusing. The latter description is our chosen parameterization. We will correct this.
>
> - In the first line of equation 7 it seems that the variational approximation q_phi for z_t only depends on x_t, but it is actually dependent also on the future x through s_t and q_meas
> - Is there a missing s_t in q_trans in the first line of (7)?
>
> Indeed, there are multiple conflicting specifications in (7) which we will address. q_trans is certainly conditioned on s_t as that is of course the entire point of s_t - influencing the transition of z.
>
> - One may even wonder if it is worth the effort: could you have used instead a deterministic s_t parameterized for example as a bidirectional LSTM with softmax output?
>
> With regards to deterministic switching variables s_t, this is exactly the approach the Deep Variational Bayes Filter took (although not parameterized by an RNN). We argue and try to demonstrate that the performance gains stem from the probabilistic treatment of the switching variable, be it Concrete or Gaussian.
>
> - To be able to detect walls the z variables basically need to learn to represent the position of the agent and encoding the information on the position of the walls in the connection to s_t.  Would you then need to train the model from scratch for any new environment?
>
> Given that the model was trained on the agents' global position, a retraining would certainly be required in any case. If trained on local measurements, e.g. every agent equipped with some distance sensors, one could imagine that s_t is learned based on locally encoded information only, supposing a clean split between local and global information in z_t. This, however, is mere speculation, what is shown in our experiment is mainly that a single switching variable generalizes over the entire maze, e.g. encoding all vertical walls. This is in contrast to the deterministic treatment where we found a single switching variable only covering parts of the maze as shown in figure 3.
>
> As multiple reviewers have raised concern about the chosen structure (inference treated before generative model), we will address this part in the hope to make the paper more coherent and readily understandable.

---

### Official Review · AnonReviewer3 · 2018-10-31
**The proposed approach is not clearly presented.**

**Rating:** 4
**Confidence:** 4

**Review:**

This paper proposes a new model for switching linear dynamical systems. The standard model and the proposed model are presented. Together with the inference procedure associated to the new model. This inference procedure is based on variational auto-encoders, which model the transition and measurement posterior distributions, which is exactly the methodological contribution of the manuscript. Experiments on three different tasks are reported, and qualitative and quantitative results (comparing with different state-of-the-art methods) are reported.

The standard model is very well described, formally and graphically, except for the dynamic model of the switching variable, and its dependence on z_t-1. The proposed model has a clear graphical representation, but its formal counterpart is a bit  more difficult to grasp, we need to reach 4.2 (after the inference procedure is discussed) to understand the main difference (the switching variable does not influence the observation model). Still, the dependency of the dynamics of s_t on z_t is not discussed.

In my opinion, another issue is the discussion of the variational inference procedure, mainly because it is unclear what additional assumptions are made. This is because the procedure does not seem to derive from the a posteriori distribution (at least it is not presented like this). Sometimes we do not know if the authors are assuming further hypothesis or if there are typos in the equations.

For instance (7) is quite problematic. Indeed, the starting point of (7) is the approximation of the a posteriori distribution q_phi(z_t|z_t-1,x_1:T,u_1:T), that is split into two parts, a transition model and an inverse measurement model. First, this split is neither well motivated nor justified: does it come from smartly using the Bayes and other probability rules? In particular, I do not understand how come, given that q_phi is not conditioned on s_t, the past measurements and control inputs can be discarded. Second, do the authors impose that this a posteriori probability is a Gaussian? Third, the variable s_t seems to be in and out at the authors discretion, which is not correct from a mathematical point of view, and critical since the interesting part of the model is exactly the existence of a switching variable and its relationship with the other latent/observed variables. Finally, if the posterior q_phi is conditioned to s_t (and I am sure it must), then the measurement model also has to be conditioned on s_t, which poses perhaps another inference problem.

Equation (10) has the same problem, in the sense that we do not understand where does it derive from, why is the chosen split justified and why the convex sum of the two distributions is the appropriate way to merge the information of the inverse measurements and the transition model.

Another difficulty is found in the generative model, when it is announced that the model uses M base matrices (but there are S possibilities for the switching variable). s_t(i) is not defined and the transition model for the switching variable is not defined. This part is difficult to understand and confusing. At the end, since we do not understand the basic assumptions of the model, it is very hard to grasp the contribution of the paper. In addition, the interpretation of the results is much harder, since we are missing an overall understanding of the proposed approach.

The numerical and quantitative results demonstrate the ability of the approach to outperform the state-of-the-art (at least for the normal distribution and on the first two tasks).

Due to the lack of discussion, motivation, justification and details of the proposed approach, I recommend this paper to be rejected and resubmitted when all these concerns will be addressed.

---

> ### Author Response · Authors · 2018-11-19
> **Addressing some of the confusion and justifications**
>
> We'd like to cast away at least some of the confusion caused by our paper in order to improve it.
>
> - For instance (7) is quite problematic [...] Third, the variable s_t seems to be in and out at the authors discretion
>
> Indeed, there are multiple conflicting specifications in (7) which we will address. q_trans is certainly conditioned on s_t.
>
> - First, this split is neither well motivated nor justified: does it come from smartly using the Bayes and other probability rules?
>
> This split will become clearer when we restructure the paper to describe the generative model first. The motivation is the following. It has been noted in prior work that in this kind of recurrent VAE, getting the reconstruction gradient through the generative transition is paramount for performance. In many works, the transition is only forced by the KL term, which is not enough. We'd like to achieve that by sharing parameters of the transition between generative and inference model. Therefore, we require one part of the inference model to be identical to the generative transition model so that it can be reused. This is q_trans. The other part needs to integrate the observations and then adjust whatever the prior/transition model predicted. Together, they constitute basic essence of a filter. Another motivation is: if our goal is to learn transition dynamics, why not reuse it in the inference model, where it may also be useful?
>
> Now, there is a loose parallel we can draw to the update step of a Kalman filter and the reason why this type of integration (Gaussian fusion) between q_trans and q_meas was chosen. This could be viewed as the update step of a (extended) Kalman filter if q_meas gave us an observation in the latent space. We say this with a big asterisk, as this is not to be understood as a principled argument, but just as way to think about.
>
> - Finally, if the posterior q_phi is conditioned to s_t (and I am sure it must), then the measurement model also has to be conditioned on s_t, which poses perhaps another inference problem.
>
> You are right, it is. However, only q_trans is conditioned on s_t, not q_meas which is only conditioned on observations (and possibly control inputs). Only together they constitute our variational approximation. Inference remains unproblematic as we factorize over time and can infer z_t and s_t in an alternating fashion.
>
> - In particular, I do not understand how come, given that q_phi is not conditioned on s_t, the past measurements and control inputs can be discarded.
>
> Note that q_meas alone constitutes only one part of the inference model, only in combination with q_trans may it infer the approximate posterior. We make the assumption of a Markovian state space system (z_t \indep x_1, .., x{t-1} | z{t-1}), meaning all information from observations/controls up to time t will be encoded in z_t and can come through q_trans. Therefore, we do not need to condition on past observations x_{<t} explicitly.
>
> - Second, do the authors impose that this a posteriori probability is a Gaussian?
>
> Yes. We multiply the resulting Gaussian pdfs of q_meas and q_trans, which results in another Gaussian pdf. This is the parameterization of our approximate posterior z_t.
>
> - Equation (10) has the same problem, in the sense that we do not understand where does it derive from, why is the chosen split justified and why the convex sum of the two distributions is the appropriate way to merge the information of the inverse measurements and the transition model.
>
> The split is again motivated as above, one part is the prior (q_trans) and shared between inference/generative model, the other takes into account external inputs (q_meas). As to the convex sum, this is less justified than the Gaussian multiplication, but an intuition is given in the paper. This allows the model to regulate whether external information is required, and it pays an increasing penalty when more information from the outside is required.
>
> - s_t(i) is not defined and the transition model for the switching variable is not defined. This part is difficult to understand and confusing.
>
> As another reviewer also remarked this, we recognize that this part was insufficiently detailed. We'll rework this part  and make the parameterization and sharing of weights more clear. Right now, we have a list of all parameterizations listed in Appendix B.3. This is simply implemented as a feed forward neural net.
>
> As multiple reviewers have raised concern about the chosen structure (inference treated before generative model), we will address this part in the hope to make the paper more coherent and readily understandable.

---

### Official Review · AnonReviewer2 · 2018-11-03
**Interesting ideas, but more justifications and comparisons necessary**

**Rating:** 6
**Confidence:** 3

**Review:**

Thank you for the detailed reply and for updating the draft

The authors have added in a sentence about the SLDS-VAE from Johnson et al and I agree that reproducing their results from the open source code is difficult. I think my concerns about similarities have been sufficiently addressed.

My main concerns about the paper still stem from the complexity of the inference procedure. Although the inference section is still a bit dense, I think the restructuring helped quite a bit. I am changing my score to a 6 to reflect the authors' efforts to improve the clarity of the paper. The discussion in the comments has been helpful in better understanding the paper but there is still room for improvement in the paper itself.
=============

Summary: The authors present an SLDS + neural network observation model for the purpose of fitting complex dynamical systems. They introduce an RNN-based inference procedure and evaluate how well this model fits various systems. (I’ll refer to the paper as SLDVBF for the rest of the review.)

Writing: The paper is well-written and explains its ideas clearly

Major Comments:
There are many similarities between SLDVBF and the SLDS-VAE model in Johnson et al [1] and I think the authors need to address them, or at least properly compare the models and justify their choices:

- The first is that the proposed latent SLDS generative models are very similar: both papers connect an SLDS with a neural network observation model. Johnson et al [1] present a slightly simpler SLDS (with no edges from z_t -> s_{t + 1} or s_t -> x_t) whereas LDVBF uses the “augmented SLDS” from Barber et al. It is unclear what exactly  z_t -> s_{t + 1} is in the LDVBF model, as there is no stated form for p(s_t | s_{t -1}, z_{t - 1}).

- When performing inference, Johnson et al use a recognition network that outputs potentials used for Kalman filtering for z_t and then do conjugate message passing for s_t. I see this as a simpler alternative to the inference algorithm proposed in SLDVBF. SLDVBF proposes relaxing the discrete random variables using Concrete distributions and using LSTMs to output potentials used in computing variational posteriors. There are few additional tricks used, such as having these networks output parameters that gate potentials from other sources. The authors state that this strategy allows reconstruction signal to backpropagate through transitions, but Johnson et al accomplish this (in theory) by backpropagating through the message passing fixed-point iteration itself. I think the authors need to better motivate the use of RNNs over the message-passing ideas presented in Johnson et al.

- Although SLDVBF provides more experiments evaluating the SLDS than Johnson, there is an overlap. Johnson et al successfully simulates dynamics in toy image systems in an image-based ball-bouncing task (in 1d, not 2d). I find that the results from SLDVBF, on their own, are not quite convincing enough to distinguish their methods from those from Johnson et al and a direct comparison is necessary.

Despite these similarities, I think this paper is a step in the right direction, though it needs to far more to differentiate it from Johnson et al. The paper draws on many ideas from recent literature for inference, and incorporating these ideas is a good start.

Minor Comments:

- Structurally, I found it odd that the authors present the inference algorithm before fully defining the generative model. I think it would be clearer if the authors provided a clear description of the model before describing variational approximations and inference strategies.
- The authors do not justify setting $\beta = 0.1$ when training the model. Is there a particular reason you need to downweight the KL term as opposed to annealing?

[1] Johnson, Matthew, et al. "Composing graphical models with neural networks for structured representations and fast inference." Advances in neural information processing systems. 2016.

---

> ### Author Response · Authors · 2018-11-19
> **Comparison to Johnson et al a valid concern**
>
> SLDS-VAE by Johnson et al. is certainly worth comparing to.  We considered a direct comparison to SLDS-VAE, however Johnson et al. restrict their evaluation of experiments to qualitative analysis, with the exception of one plot comparing the convergence of natural to standard gradient updates. We considered running the publicly available code on our own data, but our success has been limited (with regards to both training speed and convergence; the speed may be attributed to the underlying software packages like numpy/autograd and not the method in itself). More realistically, a comparison on the theoretical approach might be more fruitful (and warranted). It will certainly be referred to in our related work section where it should have been all along.
>
> - as there is no stated form for p(s_t | s{t -1}, z{t - 1})
>
> p(s_t | s{t -1}, z{t - 1}) is entirely shared with q_trans(s_t | s{t -1}, z{t - 1}). It's parameterized by a simple feed forward neural network and weights are shared. Furthermore, we have a list of all parameterizations listed in Appendix B.3.
>
> - The authors state that this strategy allows reconstruction signal to backpropagate through transitions, but Johnson et al accomplish this (in theory) by backpropagating through the message passing fixed-point iteration itself. I think the authors need to better motivate the use of RNNs over the message-passing ideas presented in Johnson et al.
>
> The key point is not just the gating but that parameters are shared between inference and generative model for p(s_t | s{t -1}, z{t - 1}). If we did neither of those things (parameter sharing + simple combination with a gate or the like), the learning signal for the generative transition would indeed be restricted to the KL term which has been shown to be insufficient. Now, in theory, this should also work, but in practice, results of those parameterizations have been underwhelming. With regards to Johnson et al, it is hard for us tell how good the learned transition actually is, as no quantitative evidence has been provided in the paper.
>
> Reasons for RNNs over message-passing are probably the same as for many others choosing to go for neural VI methods. Learning by backpropagation, being in principle more flexible with the chosen distributions, convergence issues of message-passing, in particular when combined with other model components. Even though you seem at least somewhat unconvinced, we think our empirical results are certainly an argument for our chosen approach/parameterization.
>
> - The authors do not justify setting \beta = 0.1 when training the model. Is there a particular reason you need to downweight the KL term as opposed to annealing?
>
> The chosen \beta has led to significantly better results. In general, the effect of s_t on the loss is small compared to z_t. What can be encoded depends a lot on the chosen time discretization, whether the variance of the decoder is learned and so forth. Downscaling the KL seemed a robust solution for our experiments which is highlighted by the fact that we chose the same \beta for all of them. There has been a lot of recent work[1,2] about the limitations of the ELBO and about justifying different scalings of KL or likelihood. At its core, the ELBO doesn't provide strong guidance on where information has to reside (latent space or in parameters of the generative model). Practical remedies have been put forward, the most trivial being a different scaling of the KL. This requires theoretical justification, but is out of scope for our paper. We just make use of empirically founded "tricks" for our empirical section. Our model, in general, is not affected.
>
> As multiple reviewers have raised concern about the chosen structure (inference treated before generative model), we will address this part in the hope to make the paper more coherent and readily understandable.
>
> [1] Danilo Jimenez Rezende and Fabio Viola. Taming VAEs.  https://arxiv.org/pdf/1810.00597.pdf
>
> [2] Alexander A. Alemi et al. Fixing a Broken ELBO. https://arxiv.org/abs/1711.00464

---

> > ### Comment · AnonReviewer1 · 2018-11-23
> > **Comparison to Johnson et al**
> >
> > I agree with the reviewer that a comparison with the SLDS-VAE from Johnson et al would be very interesting.
> > However, I understand the authors' challenges with running experiments modifying their the publicly available code, as I have also tried it in the past with no success.
> >
> > Perhaps a simpler comparison could be with against the model of [1], which is an extension of the work from Johnson et al (code available at https://github.com/emtiyaz/vmp-for-svae/).
> >
> > [1] Variational Message Passing with Structured Inference Networks. Wu Lin, Nicolas Hubacher, Mohammad Emtiyaz Khan, ICLR2018.

---

> > ### Comment · AnonReviewer2 · 2018-11-26
> > **Updated score and review**
> >
> > I have read your reply and have updated my score and comments up above.

---

### Meta-Review · Area_Chair1 · 2018-12-14
**Not quite not enough for acceptance**

**Confidence:** 5
**Recommendation:** Reject

**Metareview:**

The overall view of the reviewers is that the paper is not quite good enough as it stands. The reviewers also appreciates the contributions so taking the comments into account and resubmit elsewhere is encouraged.